# Bayesian estimation of community size and overlap from random subsamples

Erik K. Johnson[1]*, Daniel B. Larremore[2,3]*

**1** Department of Applied Mathematics, University of Colorado Boulder, Boulder, Colorado, United States of America, **2** Department of Computer Science, University of Colorado Boulder, Boulder, Colorado, United States of America, **3** BioFrontiers Institute, University of Colorado Boulder, Boulder, Colorado, United States of America

\* erik.k.johnson@colorado.edu (EKJ); daniel.larremore@colorado.edu (DBL)

## Abstract

Counting the number of species, items, or genes that are shared between two groups, sets, or communities is a simple calculation when sampling is complete. However, when only partial samples are available, quantifying the overlap between two communities becomes an estimation problem. Furthermore, to calculate normalized measures of $\beta$-diversity, such as the Jaccard and Sorenson-Dice indices, one must also estimate the total sizes of the communities being compared. Previous efforts to address these problems have assumed knowledge of total community sizes and then used Bayesian methods to produce unbiased estimates with quantified uncertainty. Here, we address communities of unknown size and show that this produces systematically better estimates—both in terms of central estimates and quantification of uncertainty in those estimates. We further show how to use species, item, or gene count data to refine estimates of community size in a Bayesian joint model of community size and overlap.

## Author summary

When two sets of species, genes, or items have been completely enumerated, quantifying the overlap between the sets is as simple as comparing their contents. However, in many applications, only random samples from the two sets are available, forcing the problem of overlap quantification into the realm of inference. Using a Bayesian inference approach, this paper shows how one can use random samples from two sets to simultaneously estimate the total size of each set, as well as the overlap between them. Rather than learning from the presence and absence of each species, gene, or item alone, as in prior work, this method utilizes the total number of samples drawn from each set to aid in the inference process. By drawing on this additional information, overlap estimates are more confident and accurate. These methods not only allow inference from existing data, but also enable prospective sample size calculations via simulation.

**Data Availability Statement:** All code needed to evaluate the conclusions in the paper are present in the paper and in the Supplementary Materials: S1 and S2 Text, and open-source code is available at

https://github.com/erikj540/Bayesian-Beta-Diversity/releases/tag/v1. Bayesian models were implemented in Python 3.8.

**Funding:** The work of DBL was supported in part by the SeroNet program of the National Cancer Institute (1U01CA261277-01) and by an Alan T. Waterman Award from the National Science Foundation (SMA-2226343). The funders had no role in study design, data collection and analysis, decision to publish, or preparation of the manuscript.

**Competing interests:** The authors have declared that no competing interests exist.

This is a *PLOS Computational Biology* Methods paper.

## Introduction

Quantifying the overlap between two groups, sets, or communities is a problem in many fields including genetics, ecology, and computer science. When the two communities are fully known, one can simply count the size of their intersection. However, when populations are only partially observed, due to a subsampling or stochastic sampling process, the community overlap problem becomes one of inference.

In ecology, the relationship between the diversity in one community and another is called $\beta$-diversity [1], an idea which has led to the creation of numerous indices and coefficients which seek to quantify it. For example, the canonical Jaccard index [2] and the Sorenson-Dice coefficient [3, 4] have the appealing properties that (i) they are based only on the number of shared species, $s$, and the numbers of species in each community, $R_a$ and $R_b$, and they take the values zero, when two communities are entirely unrelated, and one, when the communities are identical. However, these coefficients, as well as alternatives [5], have been shown to be biased when community sampling is incomplete [6, 7]. Furthermore, they provide no measure of statistical uncertainty because they provide only point estimates.

To address these issues, improvements in the quantification of $\beta$-diversity have been made in various ways. One direction of development recognizes that the measurement of $\beta$-diversity from the presence and absence of species fundamentally relies on counting the species shared by the two communities in the context of the numbers of species in each community separately, thus cataloguing the myriad ways in which these three integers might be reasonably combined, depending on the circumstances [5]. Another set of developments has been to work with species abundance data instead of binary presence-absence measurements [8]. A third set of developments has been to place observations of both abundance and presence-absence in the context of a probabilistic sampling process [6, 7], allowing for the appropriate quantification of uncertainty through confidence intervals or credible intervals.

One key feature of the $\beta$-diversity measures that quantify uncertainty is that the assumptions of their underlying statistical models must be stated explicitly. This provides transparency and also reveals assumptions which may not hold in practice. In recent work, a Bayesian approach to $\beta$-diversity estimation was introduced which provides unbiased estimates of the overlap between two stochastically sampled communities, yet this approach assumes that the two original community sizes are known a priori [7]. In practice, however, overall community sizes may be unknown, or may vary widely, making this model and others like it misspecified from the outset to an unknown degree. Thus, while incorporating appropriate uncertainty into community overlap estimation is an improvement, doing so without recognizing uncertainty or misspecification in each individual community's size may nevertheless lead to biased, overconfident, and unreliable inferences.

Here we address this problem by leveraging an additional and often available source of data in presence-absence studies: the total number of independent samples taken from each community, i.e. the sampling depth or effort. Building on the same intuition as the estimation of total species from a species accumulation curve [9], we introduce a model for $\beta$-diversity calculations which produces joint estimates of $s$, $R_a$, and $R_b$ in a Bayesian statistical framework. Posterior samples of these quantities offer solutions to issues identified above by providing unbiased central estimates, the quantification of uncertainty via credible intervals, and the construction of Bayesian versions of the canonical Jaccard and Sorenson-Dice coefficients (as well as 20 others which are based on $s$, $R_a$, and $R_b$ [5]).

Although estimating pairwise similarity is a problem in many fields, here we present the problem in the context of estimating the genetic similarity between pairs of malaria parasites from the species *Plasmodium falciparum*—the most virulent of the human malaria parasites. Because terminology varies by context, in the remainder of this manuscript we use the terms community, set, and repertoire to refer to the same fundamental thing: the total number of unique species, objects, or genes, respectively, in a group of interest. Our goal in all contexts will be to estimate the number of shared species, objects, or genes, and to simultaneously estimate the sizes of each of the two communities, sets, or repertoires being compared.

## *P. falciparum* repertoire overlap problem

During the blood stage of malaria, *P. falciparum* parasites replicate inside erythrocytes, and export a protein to the erythrocytic surface, called *Plasmodium falciparum* Erythrocyte Membrane Protein 1 (PfEMP-1). There, the PfEMP-1 will allow the infected erythocyte to bind to human endothelial cells, facilitating the sequestration of the infected erythrocyte away from free circulation. Due to this important role, *var* genes have been widely studied and linked to malaria's virulence and duration of infection [10–14].

Rather than a single *var* gene (and thus a single PfEMP-1), each *P. falciparum* genome contains a repertoire of hypervariable and mutually distinct *var* genes [15]. The *var* genes differ within and between parasites, due to rapid recombination and reassortment [16, 17]. This variability in *var* genes, and thus in PfEMP-1, facilitates immune evasion while preserving the ability to bind to different types of endothelial receptors. Critically, the number of *var* genes found in each parasite's repertoire varies considerably [18]. For instance, the reference parasite 3D7 has been measured to have 58 *var* genes [15] while the DD2 and RAJ116 parasites have 48 and 39, respectively [19].

Studies of *P. falciparum* epidemiology and evolution have generated insights by comparing the *var* repertoires between parasites through $\beta$-diversity calculations [20–27]. Theory suggests that if a human population has been exposed to particular *var* genes, then repertoires containing those *var* genes will have lower fitness than repertoires that are entirely unrecognized by local hosts, shaping the *var* population structure [23–25, 28–30]. Thus, these linked immunological, epidemiological, and evolutionary questions require careful consideration of the methods by which we estimate the extent to which *var* repertoires overlap. However, traditional estimates of overlap between *var* repertoires suffer bias due to subsampling, mirroring similar observations for $\beta$-diversity measures more broadly [6].

Due to the massive diversity and recombinant structure of *var* genes, *var* studies typically use degenerate PCR primers to target a small "tag" sequence within a single *var* domain called DBL$\alpha$ [31]. These DBL$\alpha$ tags are widely used to study the structure and function of *var* genes [13, 20, 23, 31–36], but due to limited resources and/or time, DBL$\alpha$ PCR data are typically a random subsample from each parasite's *var* repertoire. These PCR-based subsampling procedures therefore produce both presence-absence data for various *var* types, and counts reflecting the number of times each present *var* was observed.

In this context, repertoire overlap is typically called pairwise type sharing [20] and is often quantified by the the Sorenson-Dice coefficient:

$$\widehat{SD}_{\text{Empirical}} = \frac{n_{ab}}{\frac{1}{2}(n_a + n_b)} \tag{1}$$

where $n_a$ and $n_b$ are the number of unique *var* types sampled from parasites *a* and *b*, respectively, and $n_{ab}$ is the number of sampled types shared by both parasites (i.e., the empirical overlap). When repertoires are not fully sampled (as is overwhelmingly the case in existing studies

[20–23, 25, 26]) the Sorensen-Dice coefficient underestimates the true overlap between repertoires. Problematically, this downward bias increases as $n_a$ and $n_b$ decrease [6, 7], which prevents direct comparisons between study sites with different sampling depths.

The methods introduced in this paper, while targeted more broadly at the development of *β*-diversity quantification, are developed in the particular context of this *P. falciparum* repertoire overlap problem.

## Methods

### Setup

Our method for inferring overlap is based on two key observations. First, not all repertoires are the same size but information about a repertoire's size can be gleaned from the rate at which more samples identify new repertoire elements [9]. Second, the observed overlap $n_{ab}$ is a realization of a stochastic sampling process which depends on not only the true overlap but also the true repertoire sizes. These observations lead us to use a hierarchical Bayesian approach (Fig 1).

In brief, we model the stochastic process that generates the observed presence-absence data ($n_a$, $n_b$, and $n_{ab}$) which can be derived from observed sample counts (i.e. observed abundances, $C_a$, $C_b$), from two parasites with repertoire sizes $R_a$ and $R_b$ and overlap $s$. The core of this stochastic sampling process is the assumption that sampling from each repertoire is done independently, uniformly at random, and with replacement, corresponding to PCR of *var* gDNA without substantial primer bias. From this model, we compute the joint posterior distribution of the unknown parameters, $s$, $R_a$, and $R_b$. With this joint posterior distribution, $p(s, R_a, R_b \mid C_a, C_b)$, we can produce unbiased *a posteriori* point estimates of the repertoire sizes and overlap, and can quantify uncertainty in these point estimates via credible intervals.

In the detailed methods that follow, we describe our choice of priors over the three parameters $s$, $R_a$, and $R_b$, derive the model likelihood, and review the steps required to make calculations efficient. An open-source implementaton of these methods is freely available (see Code Availability statement).

### Choice of prior distributions

Due to extensive sequencing and assembly efforts [18], the repertoire sizes for thousands of *P. falciparum* parasites have been characterized, leading us to choose a data-informed prior distribution for repertoire sizes $R_a$ and $R_b$. We assume an informative Poisson prior for $R_a$ and $R_b$, fit to the repertoire sizes from 2398 parasite isolates published by Otto et al. [18].

$$R_a, R_b \sim \mathrm{Poisson}[55].$$

For *β*-diversity studies outside of *P. falciparum*, alternative informative priors can be chosen. Because the repertoire overlap $s$ can take values between 0 and $\min\{R_a, R_b\}$, we use an uninformative prior for repertoire overlap $s$,

$$s \mid R_a, R_b \sim \mathrm{Uniform}\ [0, \min\ \{R_a, R_b\}]\ .$$

### Computing the joint posterior distribution $p(s, R_a, R_b \mid C_a, C_b)$

The posterior distribution of the parameters given the count data is a product of three terms

$$p(s, R_a, R_b \mid C_a, C_b) = p(s \mid n_a, n_b, n_{ab}, R_a, R_b) \cdot p(R_a \mid C_a) \cdot p(R_b \mid C_b)\ , \tag{2}$$

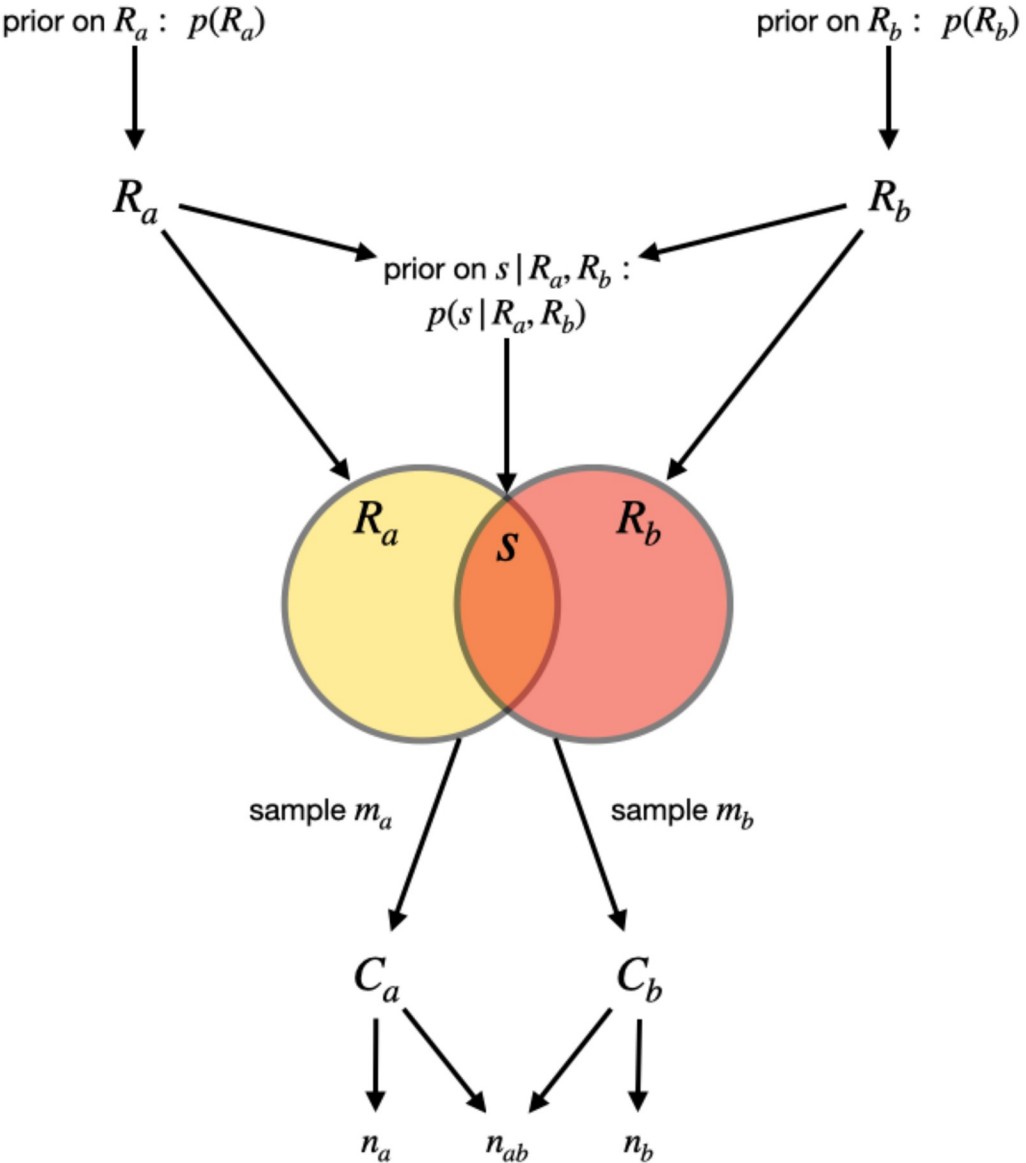

**Fig 1. Diagram of the model.** Two repertoire sizes, $R_a$ and $R_b$, are generated by their priors. The overlap between the repertoires, $s$, is then generated by the prior on the overlap given the repertoire sizes. The repertoire sizes and overlap define the two parasites, $a$ and $b$, from which we sample. Sampling $m_a$ items with replacement from parasite $a$ produces count data $C_a$ consisting of genes sampled from parasite $a$ and counts per gene. Sampling $m_b$ items with replacement from parasite $b$ produces count data $C_b$ consisting of genes sampled from parasite $b$ and counts per gene.

a calculation shown in detail in S1 Text. The rest of this section is devoted to computing each of these terms, noting that that last two are mathematically identical, but derived from different data.

To compute $p(R \mid C)$, the distribution of repertoire size given count data for a fixed but arbitrary total sampling effort $m_a = m_b = m$, we first calculate the likelihood of observing count data $C$ given a repertoire size $R$, i.e., $p(C \mid R)$. Knowing how to compute $p(C \mid R)$, allows us to

calculate $p(R \mid C)$ via Bayes' rule

$$p(R \mid C) = \frac{p(C \mid R) \cdot p(R)}{p(C)} = \frac{p(C \mid R) \cdot p(R)}{\sum_{R_i} p(C \mid R_i) \cdot p(R_i)} \tag{3}$$

where $p(R)$ is the prior on repertoire size and the sum in the denominator should be computed over the support of $p(R)$. For the unbounded support of the Poisson prior used here, we restrict the sum to only those terms above the numerical precision of the computer.

In S2 Text, we prove that

$$p(C \mid R) = \frac{R!}{(R - n)! \cdot f_1! \cdot f_2! \cdots f_Q!} \cdot \frac{m!}{c_1! \cdot c_2! \cdots c_n!} \cdot \frac{1}{R^m} \tag{4}$$

where the $c_i$ are the number of times each of the $n$ sampled *var* types were observed and the $f_i$ are the multiplicities of the unique numbers in $\{c_i\}_{i=1}^n$. For instance, suppose the count data consists of five unique *var* types with counts

$$\{c_1, c_2, c_3, c_4, c_5\} = \{1, 1, 2, 2, 3\} \tag{5}$$

then there are three ($Q = 3$) unique numbers amongst the $c_i$: 1, 2, and 3. Further, 1's multiplicity in $\{1, 1, 2, 2, 3\}$ is 2, 2's is 2, and 3's is 1 so $(f_1, f_2, f_3) = (2, 2, 1)$.

With the likelihood $p(C \mid R)$ in hand, it is straightforward to calculate the posterior $p(R \mid C)$ via Eq (3). And, thus, we can calculate the second and third terms in Eq (2).

Conveniently, the remaining term of Eq (2) $p(s \mid n_a, n_b, n_{ab}, R_a, R_b)$ has been derived in the literature [7], but only under the restriction that $R_a = R_b = 60$. We therefore rederive this quantity for general but fixed $R_a$ and $R_b$, summarizing the main steps here.

Using Bayes' rule, we can write

$$p(s \mid n_a, n_b, n_{ab}, R_a, R_b) \propto p(n_{ab} \mid n_a, n_b, s, R_a, R_b) \cdot p(s \mid R_a, R_b) \tag{6}$$

where $p(s \mid R_a, R_b)$ is a user-specified prior described above. The other term, $p(n_{ab} \mid n_a, n_b, s, R_a, R_b)$, can be computed by considering the probability that two subsets of size $n_a$ and $n_b$ will have an intersection of size $n_{ab}$, given that they have been drawn uniformly from sets of total size $R_a$ and $R_b$ whose intersection is size $s$. To do so, we use the hypergeometric distribution, $\mathcal{H}(s, R, n)$, which is the distribution of the number of "special" objects drawn after $n$ uniform draws with replacement from a set of $R$ objects, $s$ of which are "special." With this distribution in mind, note that observing $n_{ab}$ shared *var* genes can be thought of as a two-step process. First, draw $n_a$ *var* genes from parasite $a$'s $R_a$ total in which $s$ are special because they are shared with parasite $b$. The number of shared *var*s drawn is a random variable $s_a \sim \mathcal{H}(s, R_a, n_a)$. Second, draw $n_b$ genes from parasite $b$'s $R_b$ total in which $s_a$ are special because they are shared by both parasites *and* were drawn from parasite $a$. The number of shared *var*s captured after sampling from both parasites, $n_{ab}$, will be distributed according to $\mathcal{H}(s_a, R_b, n_b) = \mathcal{H}(\mathcal{H}(s, R_a, n_a), R_b, n_b)$.

To generate a particular empirical overlap $n_{ab}$, first step 1 must happen and then, independently, step 2 must happen. We therefore multiply these two hypergeometric probabilities. However, because these two steps may occur for any value of the intermediate variable $s_a$, we

sum over all possible values of $s_a$

$$p(n_{ab} \mid n_a, n_b, s, R_a, R_b) = \sum_{s_a=0}^{s} p(s_a \mid n_a, R_a, s) \cdot p(n_{ab} \mid n_b, R_b, s_a) \tag{7}$$

$$= \sum_{s_a=0}^{s} p(\mathcal{H}(s, R_a, n_a) = s_a) \cdot p(\mathcal{H}(s_a, R_b, n_b) = n_{ab}) \tag{8}$$

Plugging this into Eq (6) allows us to compute $p(s \mid n_a, n_b, n_{ab}, R_a, R_b)$.

## Inference method summary

We now have all the pieces in place to compute $p(s, R_a, R_b \mid C_a, C_b)$:

$$
\begin{aligned}
p(s, R_a, R_b \mid C_a, C_b) \propto{} & p(R_a) \cdot p(R_b) \cdot p(s \mid R_a, R_b) \\
& \cdot \left[ \sum_{s_a=0}^{s} p\ (\mathcal{H}(s, R_a, n_a) = s_a) \cdot p(\mathcal{H}(s_a, R_b, n_b) = n_{ab}) \right] \\
& \times \left[ \frac{R_a!}{f_1^a! \cdot f_2^a! \cdots f_{Q_a}^a!} \cdot \frac{m_a!}{c_1^a! \cdot c_2^a! \cdots c_{R_a}^a!} \left( \frac{1}{R_a} \right)^{m_a} \right] \\
& \times \left[ \frac{R_b!}{f_1^b! \cdot f_2^b! \cdots f_{Q_b}^b!} \cdot \frac{m_b!}{c_1^b! \cdot c_2^b! \cdots c_{R_b}^b!} \left( \frac{1}{R_b} \right)^{m_b} \right]
\end{aligned}
\tag{9}
$$

where the first three terms are the user-specified priors. With this joint posterior distribution, we can compute unbiased Bayesian estimates of $s$, $R_a$, and $R_b$ as expectations over the posterior:

$$\widehat{s} = \sum_{s, R_a, R_b} s \cdot p(s, R_a, R_b \mid C_a, C_b) \tag{10}$$

$$\widehat{R_a} = \sum_{s, R_a, R_b} R_a \cdot p(s, R_a, R_b \mid C_a, C_b) \tag{11}$$

$$\widehat{R_b} = \sum_{s, R_a, R_b} R_b \cdot p(s, R_a, R_b \mid C_a, C_b) \tag{12}$$

Moreover, and importantly, we can compute unbiased Bayesian estimates of any functional combination of $s$, $R_a$, and $R_b$ such as Bayesian versions of the Jaccard index [2], the Sorensen-Dice coefficient [4], other coefficients based on $s$, $R_a$, and $R_b$ [5], and the directional pairwise-type-sharing measures of He et al. [29]. For all of these measures, in addition to the point estimates, the ability to draw from the joint posterior distribution Eq (9) enables one to compute credible intervals to quantify uncertainty.

## Generation of simulated data

To facilitate numerical experiments in which we tested our inference method's ability to recover accurate estimates of $s$, $R_a$, and $R_b$, we generated synthetic data via simulation as follows. First, we selected a value of overlap $s$ between 0 and 70, so that analyses could be stratified according to overlap. Next, we drew repertoire sizes $R_a$ and $R_b$ independently from the prior distribution, ensuring that $R_a \leq s$ and $R_b \leq s$, redrawing as necessary. Next, we drew from the

model (Fig 1) a set of $m_a$ and $m_b$ samples from repertoires of sizes $R_a$ and $R_b$, respectively, with specified overlap $s$, to generate count data histograms $C_a$ and $C_b$. This procedure therefore stochastically created synthetic count data for a specified overlap $s$ and sampling depth $m_a = m_b = m$, allowing us to test our method's accuracy and uncertainty quantification under various scenarios.

## Results

### Inference

We first investigated how increasing the total number of independent samples improves our ability to correctly estimate the total size of a single repertoire (or generally, community), by which we specifically mean the number of unique constituent genes (or generically, species or objects). To do so, we conducted numerical experiments where we presumed a repertoire size and then simulated samples from it to produce count data. An example of such an experiment shows how posterior estimates approach the true repertoire size as sampling effort increases (Fig 2). Here, because we focus on a single repertoire in isolation, we drop $a$ and $b$ subscripts for the moment, referring to simply sampling effort $m$, repertoire size $R$, and count data $C$.

This experiment illustrates two related points. First, there is valuable information in knowing the total sampling effort $m$, even if some samples were duplicate observations of previously observed genes, simply because those sample counts inform repertoire size estimates. Second, increasing the sampling effort concentrates $p(R \mid C)$ around the true repertoire size, concretely linking sampling effort to estimation of not only repertoire size, but through decreased uncertainty, eventual overlap estimates as well.

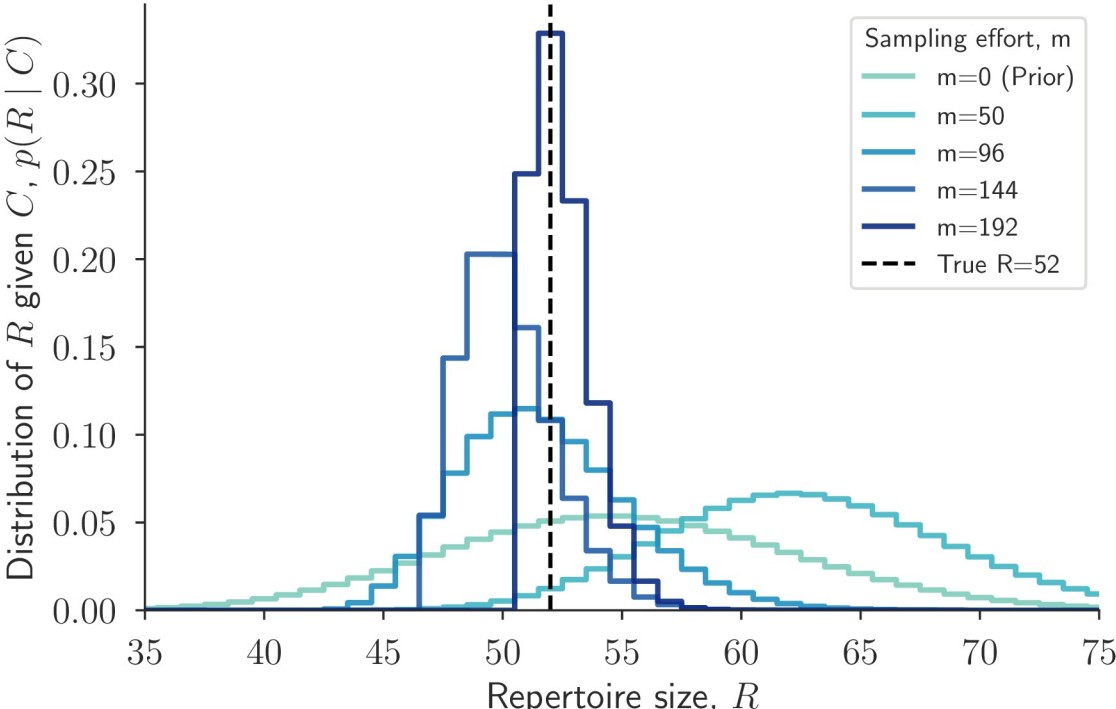

**Fig 2. Repertoire size posterior estimates improve with increased sampling effort.** For a single repertoire with true size $R = 52$, the posterior distribution $p(R \mid C)$ is plotted for different sampling efforts $m$ (see legend). For each value of $m$, count data $C$ were generated by drawing $m$ genes uniformly with replacement from a repertoire of 52 genes. As sampling effort increases, the posterior $p(R \mid C)$ concentrates around the true repertoire size 52. The $m = 0$ curve is the Poisson prior on repertoire size, $p(R)$.

Next we examined whether the $\widehat{s}$, $\widehat{R}_a$, and $\widehat{R}_b$ estimates in Eqs (10)–(12) are accurate across a range of sampling efforts $m$ in two steps. First, we simulated the sampling process for various values of $s$, $R_a$, and $R_b$ to produce synthetic count data $C_a$ and $C_b$ with varying levels of overlap between the observed samples. Then, we evaluated our ability to recover $s$, $R_a$, and $R_b$ by applying Eqs (10)–(12) to the synthetic data.

We found that the overlap and repertoire estimates accurately reproduce the true parameter values, provided that sampling effort is sufficiently large. Furthermore, as sampling effort increases, estimates become increasingly accurate (Fig 3).

However, we also observed that when the sampling effort is small but repertoires are large and highly overlapping (e.g. $m = 50$ and $s > 50$), $\widehat{s}$ underestimates the true values (Fig 3A). This phenomenon is due to a more general property of Bayesian inference: when there are fewer samples from which to infer, the prior distribution exerts a stronger effect on inferences. Here, the Poisson prior over repertoire sizes assigns low probability to repertoire sizes as large as 70 ($p(R_a \geq 70) = 0.03$), and thus, in the absence of a large sampling effort to overwhelm that prior, the surprisingly large repertoire sizes and overlaps require substantially more samples to establish. In real data from *P. falciparum*, repertoires (and thus repertoire overlaps) larger than 60 are rarely observed [18, 26], decreasing the potential impact of this issue for the study of repertoire overlap between individual parasites (though not for the study of overlap between infections containing multiple parasites; see Discussion.

## Uncertainty

Bayesian methods also allow us to quantify uncertainty via credible intervals (CIs). To measure how well our CIs capture the true parameter values, we computed 95% highest density posterior intervals for parameter estimates in simulated data, where true values were known. As expected, uncertainty decreased as sampling effort increased, and approximately 95% of the 95% CIs captured the true parameter values, as designed (Fig 4). For instance, for sampling efforts of $m = 50$, $m = 96$, and $m = 192$, the proportions of the 95% $\widehat{s}$ CIs containing the true $s$

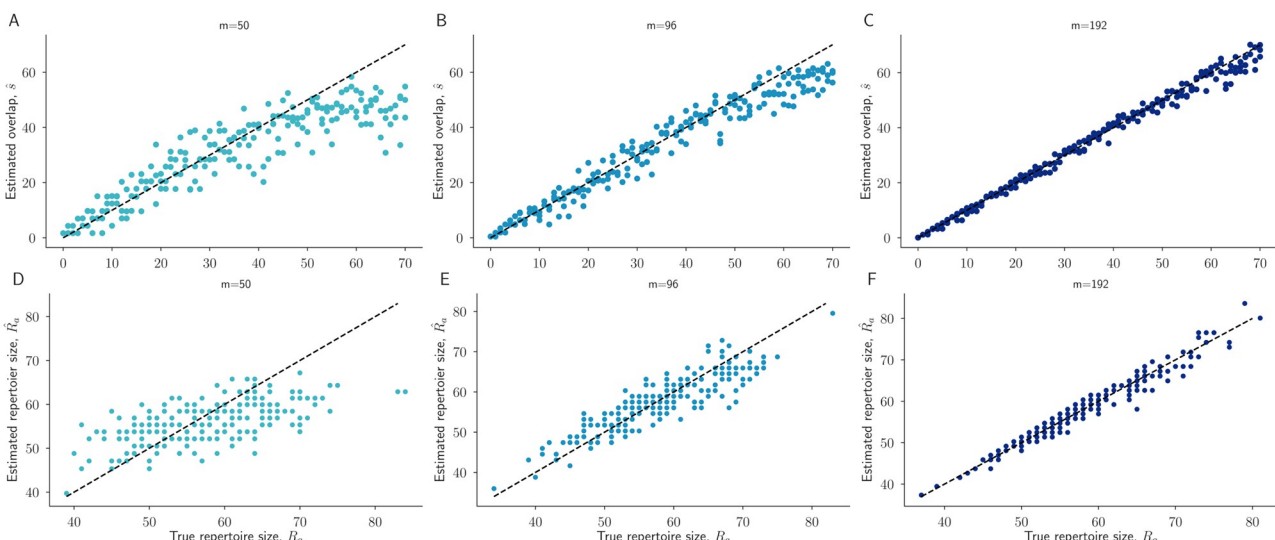

**Fig 3. Accuracy of estimates across a range of true parameter values and sampling efforts.** For each overlap value $s$ between 0 and 70, we performed three independent simulations to generate synthetic count data (Methods). Estimates of $s$ (A,B,C) and $R_a$ (D,E,F) from the resulting count data, using our statistical model, are shown. Estimates are shown for sampling efforts $m_a = m_b = m = 50$, 96, 192 across left, middle, and right columns, respectively. Dashed black lines represent perfect unbiased inference.

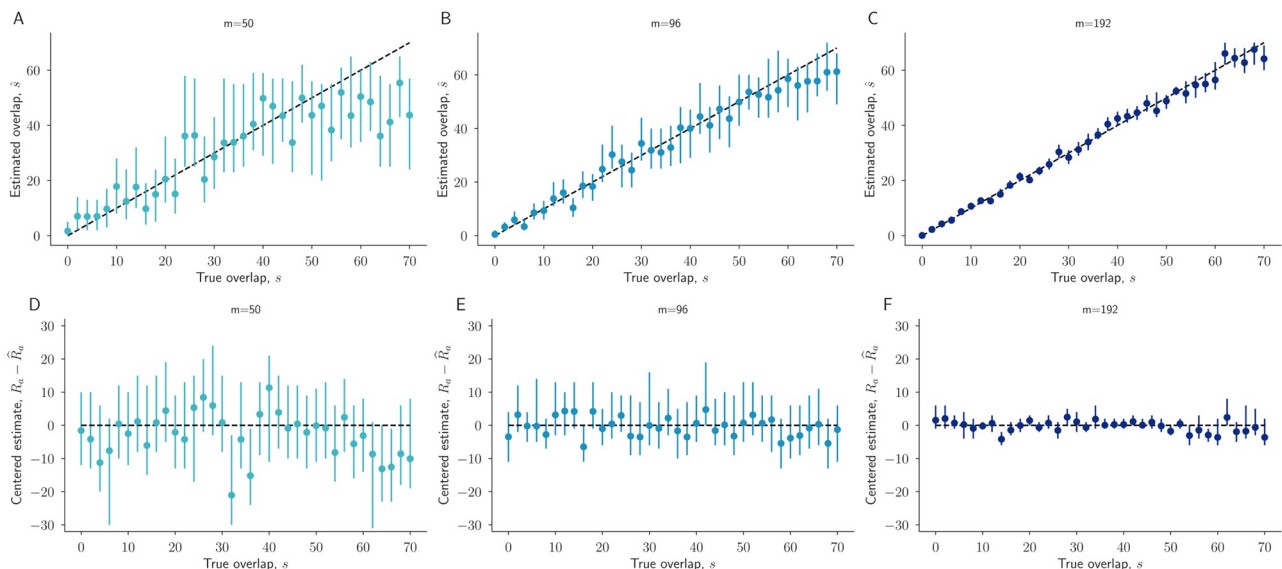

**Fig 4. Credible intervals quantify uncertainty in overlap estimates.** For each overlap value $s$ between 0 and 70, we performed one simulation to generate synthetic count data (Methods). Estimates from the resulting count data, using our statistical model, of $s$ (A,B,C), and error in $R_a$ and $R_b$ (D,E, F) are shown. Estimates (dots) and 95% credible intervals (lines) are shown for sampling efforts $m$ = 50, 96, 192 in left, middle, and right columns, respectively.

were 0.975, 0.975, and 0.965, respectively. For the same three sampling efforts, the proportions of the 95% $\widehat{R}_a$ CIs that contained the true repertoire size $R_a$ were 0.920, 0.950, and 0.955, respectively.

## Improving $\beta$-diversity indices

Over 20 different indices of $\beta$ diversity have been proposed which algebraically combine empirical estimates of $R_a$, $R_b$, and $s$ [5], including the well known Jaccard index and the Sorenson-Dice coefficient. The Sorenson-Dice coefficient is defined as the ratio of repertoire overlap to the average of the repertoires sizes,

$$SD = \frac{s}{\frac{1}{2}\left(R_a + R_b\right)} \quad . \tag{13}$$

Typically, in the absence of more sophisticated estimates of $R_a$, $R_b$, and $s$, empirical values are used,

$$\widehat{SD}_{\text{Empirical}} = \frac{n_{ab}}{\frac{1}{2}\left(n_a + n_b\right)} \tag{14}$$

However, the joint posterior distribution Eq (9) over $s$, $R_a$, and $R_b$ opens the door to a Bayesian reformulation of the Sorenson-Dice coefficient as

$$\widehat{SD}_{\text{Bayesian}} = \sum_{s,R_a,R_b} \frac{s}{\frac{1}{2}\left(R_a + R_b\right)} \cdot p(s, R_a, R_b \mid C_a, C_b) \tag{15}$$

with similar generalizations for the Jaccard coefficient or other combinations of $s$, $R_a$, and $R_b$

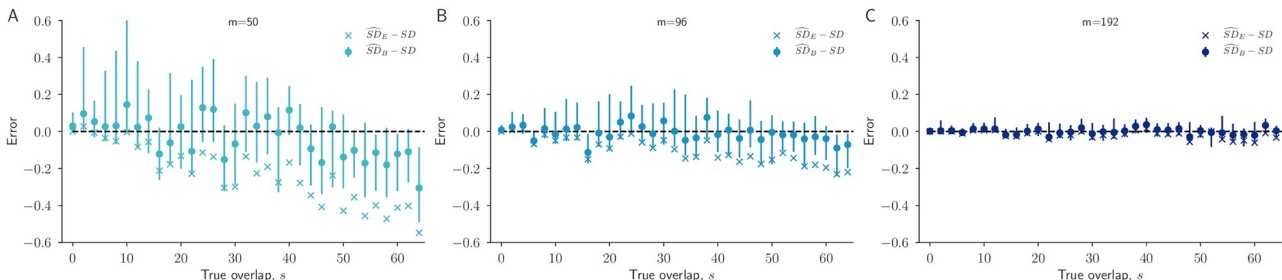

**Fig 5. Bayesian vs empirical Sorensen-Dice estimates.** For each overlap value *s* between 0 and 70, we performed one independent simulation to generate synthetic count data (Methods) and estimated the Sorensen-Dice coefficient using estimates from our Bayesian framework as well as from the raw empirical data. The error in the Bayesian Sorensen-Dice estimate, $\widehat{SD}_B$ (Eq (15)), and accompanying 95% credible intervals are shown. The often-used empirical Sorensen-Dice estimate, $\widehat{SD}_E$ (Eq (14)), is also shown. The dashed black line at 0 represents the true Sorensen-Dice coefficient (Eq (13)).

[5]. This Bayesian Sorenson-Dice coefficient averages the values of the typical Sorenson-Dice coefficient over joint posterior estimates of *s*, $R_a$, and $R_b$.

We investigated the performance of the Bayesian Sorenson-Dice coefficient $\widehat{SD}_{\text{Bayesian}}$ and its empirical counterpart $\widehat{SD}_{\text{Empirical}}$ by once more simulating the sampling process under known conditions and applying both formulas. As in our estimates of repertoire overlap, we again found that Bayesian Sorenson-Dice estimates produce consistent and unbiased estimates with correct quantification of uncertainty via credible intervals (Fig 5), except when sampling effort is low (*m* = 50) while true repertoire overlap is extremely high (*s* > 50). Furthermore, the Bayesian estimates track the true Sorenson-Dice values better than direct empirical estimates across overlap values and sampling efforts; direct empirical estimates are biased more and more downward as sampling effort decreases and as true overlap increases (Fig 5). While this illustrates how the Bayesian framework herein may be used to improve classical and commonly used estimators via Eq (15), an identical approach may be used to compute Bayesian Jaccard coefficients, or other algebraic combinations of *s*, $R_a$, and $R_b$ [5].

## Sample size calculations

Sample size calculations ask how many samples are needed to produce eventual estimates with a pre-specified level of (or upper bound on) statistical uncertainty. Such questions, while critical in the ethical study of human subjects, are also important when budgeting for studies in which additional samples require time, reagents, and funding.

To assist in sample size calculations, we used simulations to quantify the relationship between increases in sampling effort and decreases in the typical width of the credible interval around the repertoire overlap estimate estimate $\widehat{s}$ (Eq (10)). For many overlap-sampling effort pairs, (*s*, *m*), we performed 300 independent replicates in which we generated synthetic data, computed the posterior distribution for *s*, and calculated the width of the 95% $\widehat{s}$ CI.

We found that, as expected, increased sampling effort leads to decreased uncertainty across all values of overlap *s* (Fig 6). However, we also found that overlap plays a role as well, with larger overlap causing wider CIs. For instance, after *m* = 200 samples, a CI for overlap *s* = 70 is typically of width 8, while a CI for overlap *s* = 30 is typically of width 4. After *m* = 300 samples from each repertoire, median CI widths are 4 or lower for all overlap values. In short, it is easier to show with high confidence that two samples do not overlap than to show that they are highly overlapping.

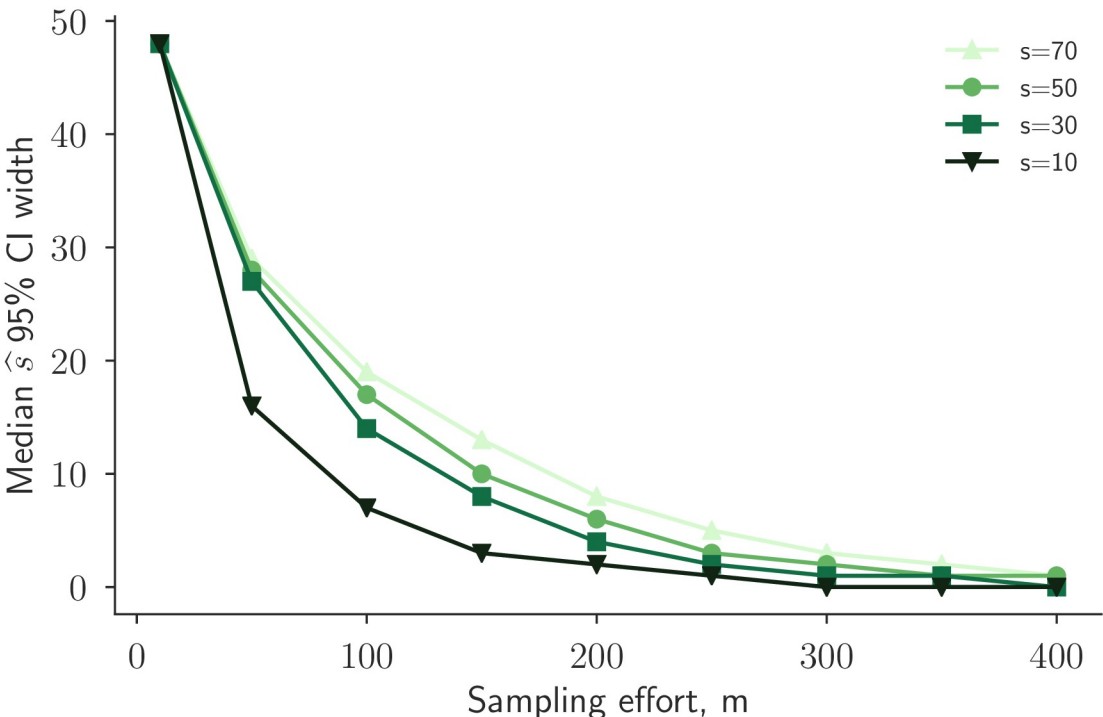

**Fig 6. Quantifying the decrease in uncertainty from increased sequencing.** Constant *s* curves show the median 95% credible interval (CI) width for the *s* estimate, $\hat{s}$, as a function of the sampling effort $m_a = m_b = m$. For each $(s, m)$-duplet, the median is across 300 count data generation simulations. This plot illustrates the intuition that additional laboratory efforts (increasing $m$) lead to higher accuracy (smaller CIs).

## Discussion

This manuscript presents a Bayesian solution to estimating the overlap between two communities, repertoires, or sets, when only subsamples are available. Importantly, because the total community sizes bear on the inference of overlap, this method jointly estimates community sizes and overlap from the quantitative accumulation of evidence, improving inferences. Samples from the joint posterior distribution can be used to quantify uncertainty via credible intervals, or can be used in Bayesian versions of the Jaccard index, Sorenson-Dice coefficient, and other algebraic combinations of set sizes and intersections. By showing how the inclusion of total sampling effort can improve inferences, this study demonstrates the value of recording and reporting not only presence-absence, but abundance as well—even when the true abundances are uniformly equal, as in the study of *P. falciparum*'s *var* gene families.

In addition to the analysis of existing data, this approach can also be used prospectively to perform sample size calculations. Importantly, context-specific sample sizes can be estimated by including additional information in the Bayesian prior. For instance, in the context of malaria's *var* genes, it is known that parasites from South America tend to have smaller repertoires [37, 38] than samples from other regions [18]—information which can be expressed through the prior distribution to influence (and in this case, decrease) sampling needs. Because additional sampling has financial and complexity costs, this allows researchers to weigh accuracy requirements against laboratory costs in the contexts of a particular study.

Beyond the study of *P. falciparum*, the approach introduced in this work lands in between two existing classes of *β*-diversity measures in the ecology literature. One class of methods measures *β*-diversity in terms of species presence or absence [5], while the other further

includes species abundance [6]. The present work uses abundance measurements (which we call count data) in order to improve presence-absence-based $\beta$-diversity estimates, but does not construct abundance-based similarity measures per se. By drawing inferences from both, this work also aligns with past efforts which rely in principle on an idea that one may draw inferences both from what is observed and what is not observed [6, 7].

The tradeoffs for improved inferences are twofold. First, our approach requires abundance data (i.e., count data $C$) instead of presence/absence totals $n_a$, $n_b$, and $n_{ab}$. This limits the retrospective analysis of past work or meta-analyses to only those studies that meet a greater data-sharing burden. However, we also note that, as proven in S2 Text, full count data are not necessary: the posterior $p(s, R_a, R_b \mid C_a, C_b)$ can still be computed exactly when only the sampling efforts ($m_a$ and $m_b$) and the presence/absence values ($n_a$, $n_b$, and $n_{ab}$) are known.

The second tradeoff for improved inference is that one must specify a prior distribution for the total community sizes. In the case of the *var* gene repertoires of *P. falciparum*, data-informed prior distributions can be created for both global [18] or local [38] estimates. In this light, one may view past work on Bayesian methods for repertoire overlap [7, 24] as specifying point priors at a particular fixed repertoire size. In general, the choice of an appropriate prior is left to the user, which may require users to make explicit their prior beliefs about community size.

There are limitations to our approach which relate to our assumptions about the sampling process which generates the count data. Specifically, we have assumed throughout this work that each time a new sample is generated, this sample is drawn independently and uniformly from a population in which unique genes, species, or objects are identically represented. Thus, unlike abundance based measures [6] which assume that some species are more likely to be sampled than others, we assumed each species' selection is equiprobable. In the sampling of *var* gene sequences, for instance, methodological artifacts such as PCR primer bias may cause non-uniform sampling. One avenue for future work could be to extend our rigorous probabilistic modeling to the non-uniform sampling regime.

Another limitation, particularly for the study of *P. falciparum*, is that bulk sequencing methods may sample from multiple distinct parasite genomes when an individual's multiplicity of infection (MOI) is greater than one. Unfortunately, even if MOI is known, it is unclear how one should alter the prior $P(R)$ for samples from that individual, due to the fact that the two or more parasite genomes within a single host may, themselves, be overlapping to an unspecified degree. This may be possible to address with further assumptions and associated priors in future work, but as a consequence, the methods presented here are valid for the analysis of *P. falciparum* only when MOI equals one.

## Supporting information

**S1 Text. Factorization of the joint posterior distribution.**
(PDF)

**S2 Text. Theorems enabling efficient computations.**
(PDF)

## Acknowledgments

The authors wish to thank Shazia Ruybal-Pesantez, Kathryn Tiedje, Karen Day, and Thomas Otto for the generosity of their feedback.

## Ethics declaration

E.K.J. and D.B.L. declare no competing interests.

## Author Contributions

**Conceptualization:** Erik K. Johnson, Daniel B. Larremore.

**Data curation:** Erik K. Johnson.

**Formal analysis:** Erik K. Johnson, Daniel B. Larremore.

**Funding acquisition:** Daniel B. Larremore.

**Investigation:** Erik K. Johnson, Daniel B. Larremore.

**Methodology:** Erik K. Johnson, Daniel B. Larremore.

**Project administration:** Daniel B. Larremore.

**Software:** Erik K. Johnson.

**Supervision:** Daniel B. Larremore.

**Validation:** Erik K. Johnson, Daniel B. Larremore.

**Visualization:** Erik K. Johnson, Daniel B. Larremore.

**Writing – original draft:** Erik K. Johnson, Daniel B. Larremore.

**Writing – review & editing:** Erik K. Johnson, Daniel B. Larremore.

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
