## [Decision Letter · Decision Letter 0]

25 Oct 2021

Dear Dr. Larremore,

Thank you very much for submitting your manuscript "Bayesian estimation of population size and overlap from random subsamples" for consideration at PLOS Computational Biology.

As with all papers reviewed by the journal, your manuscript was reviewed by members of the editorial board and by several independent reviewers. In light of the reviews (below this email), we would like to invite the resubmission of a significantly-revised version that takes into account the reviewers' comments.

Both the reviewers are overall very positive about the manuscript. One important point, raised by Reviewer #1, concerns the overlap of material with Larremore, Plos CB (2019). Some sentences are perfectly overlapping (e.g. "Of the diverse multigene families of P. falciparum, the var family is the most heavily studied because of its direct links to both malaria’s duration of infection and its virulence "). The overlap seems to be limited to introductory parts, and therefore do not affect the originality of the results. I suggest however to rephrase those parts that are strongly overlapping with the previous paper.

More importantly, as raised by both the reviewers, the code should be made available at review stage.

We cannot make any decision about publication until we have seen the revised manuscript and your response to the reviewers' comments. Your revised manuscript is also likely to be sent to reviewers for further evaluation.

Sincerely,

Jacopo Grilli

Associate Editor

PLOS Computational Biology

Nina Fefferman

Deputy Editor

PLOS Computational Biology

Both the reviewers are overall very positive about the manuscript. One important point, raised by Reviewer #1, concerns the overlap of material with Larremore, Plos CB (2019). Some sentences are perfectly overlapping (e.g. "Of the diverse multigene families of P. falciparum, the var family is the most heavily studied because of its direct links to both malaria’s duration of infection and its virulence "). The overlap seems to be limited to introductory parts, and therefore do not affect the originality of the results. I suggest however to rephrase those parts that are strongly overlapping with the previous paper.

More importantly, as raised by both the reviewers, the code should be made available at review stage.

Reviewer's Responses to Questions

**Comments to the Authors:**

Reviewer #1: This study looks at the issue of estimating the overlap between two sets when only partial samples are available from each, and - unlike previous work (Larremore 2019) - when the true repertoire sizes of both sets are unknown. The authors take a statistical modelling approach, starting by constructing a generative model and then describing how this can be used to calculate likelihoods and perform Bayesian inference. The final method is demonstrated to have nice properties, including reducing bias in cases where there is sufficient data to overwhelm the prior, and of course having a measure of uncertainty that is missing from traditional point estimates.

Overall I think this is a strong paper. The method is very clearly described, and the general gist of the method could be followed even by those without a strong statistical background. There are clear advantages to the Bayesian formulation of the problem, and the extension presented here opens the previously described method up to a wider class of problems. The general nature of this problem also means it could find application in many areas.

Other than some very minor points (see below), my only major criticism is around code availability. It is stated in the paper that an open-source implementation of the methods is freely available, and we’re pointed to the code availability statement, which points us to a GitHub repository, which appears to be basically empty (I could just see a LICENSE, an empty README and a .gitignore at time of review). Given the great effort the authors have gone to to ensure there are no unnecessary mathematical obstacles to the user it seems a shame not to have a nicely packaged tool that can be used out of the box.

There is some pretty heavy re-use of material from the earlier 2019 paper (e.g. the P. falciparum section). I didn't calculate the overlap between these two sets of text, but in some cases I suspect it's upwards of 90%, depending on the estimation method used. I leave it to the editor to decide how big of a problem this is for this particular journal.

My recommendation is to accept with minor changes, which include the points below and also tidying up the code availability.

Minor points:

1. In the description immediately before formula (A6), consider using j to index the second series (i.e. “let f_j be the number of times u_j appears in C”). Otherwise it gives the sense that f_i somehow corresponds to c_i, when in fact these are different indices.

2. Typo: In formula (A7) a central dot is used as shorthand for multiplication, and also to mean the sequence of all values from c_2 to c_n. I think the latter should be a centre-justified ellipsis.

3. The derivation of the formula p(R | n), i.e. using just the number of unique groups and not complete counts, is mathematically correct but overly complicated and probably unnecessary. The distribution of unique items in a multinomial sample is a reasonably standard expression that has been described before, see for example (https://arxiv.org/abs/1602.05822), section “The distribution of unique items”. Citing this common result would simplify the derivation here and avoid accidentally taking credit for past work.

4. In the “Inference” section, there is the statement “The true value of R is always contained within the inferred distributions”. This might be picky, but I don’t really like this statement. It’s not clear what “contained within” really means here, as even something in the 99.99th centile is contained within the distribution. Something along the lines of “confidence increases with increasing m” seems more appropriate.

5. Grammatical mistake in the line “By drawing inferences both from this work also aligns with past efforts “.

Reviewer #2: Johnson & Larremore proposed a new Bayesian estimation of beta-diversity that jointly estimates population sizes and the overlap. They showed that the estimates are unbiased when sampling efforts are large enough compared to the population sizes. This is an extension of Larremore (2019) PloS Comp Bio where population sizes are a fixed known quantity.

In general, I think the paper is clearly written, including the results. I still have several questions though regarding the definition and the scope of application.

1) It seems that population size and repertoire size are used interchangeably in the article when describing R. However, what R really refers to is not population size, but the number of species in the population (as defined in the introduction). In ecology and evolution, population size has a very different meaning—usually in the order of 10^4-10^7. Clearly, in this article, R really refers to different species/genes per population. I think the authors should be consistent in their definition of R, and make it relevant to ecologists.

2) It would be nice if the proposed measure can be applied with data from published papers to check how SD_bayesian performs, and whether it changes any conclusions from the studies. It would be better if it’s applied to ecological data where beta-diversity is calculated.

3) On page 8, the authors stated that “In real data from P. falciparum , repertoires (and thus repertoire overlaps) larger than 60 are rarely observed [28, 15], decreasing the potential impact of this issue”. However, many patients contain multiple infections of parasites, where the total isolate size is much larger than 60. It is worthwhile to discuss how that would influence the overlap estimation when the number of infections per isolate is unknown.

4) R = Ra+Rb, and m = ma+mb, or m = ma=mb? It is not clear to me from the manuscript, how the difference between Ra, Rb or ma/mb would influence the results.

**Have the authors made all data and (if applicable) computational code underlying the findings in their manuscript fully available?**

Reviewer #1: **No: **Code availability points to a GitHub repository, which is empty at time of review and hence the code is not made available with the paper. This may be a simple mistake, for example updating the repository on a local branch but failing to push to main.

Reviewer #2: **No: **The link to the github repo does not contain any codes yet

PLOS authors have the option to publish the peer review history of their article (what does this mean?). If published, this will include your full peer review and any attached files.

Reviewer #1: No

Reviewer #2: No
---

## [Decision Letter · Decision Letter 1]

28 Jul 2022

Dear Dr. Larremore,

We are pleased to inform you that your manuscript 'Bayesian estimation of community size and overlap from random subsamples' has been provisionally accepted for publication in PLOS Computational Biology.

Before your manuscript can be formally accepted you will need to complete some formatting changes, which you will receive in a follow up email. A member of our team will be in touch with a set of requests. Please note also that reviewer 1 noted an incorrect link to GitHub; please fix this.

Best regards,

Jacopo Grilli

Associate Editor

PLOS Computational Biology

Nina Fefferman

Deputy Editor

PLOS Computational Biology

Jason A. Papin

Editor-in-Chief

PLOS Computational Biology

Reviewer's Responses to Questions

**Comments to the Authors:**

Reviewer #1: I thank the authors for taking time to make the necessary changes to the paper and code. I am satisfied with all changes, except for one mistake - the github link in code availability points to the wrong repos (vaccine efficacy rather than beta diversity).

Reviewer #2: The authors have addressed all my concerns from my previous review comments.

**Have the authors made all data and (if applicable) computational code underlying the findings in their manuscript fully available?**

Reviewer #1: Yes

Reviewer #2: Yes

PLOS authors have the option to publish the peer review history of their article (what does this mean?). If published, this will include your full peer review and any attached files.

Reviewer #1: No

Reviewer #2: No

---

## [Editor Report · Acceptance letter]

25 Aug 2022

PCOMPBIOL-D-21-01287R1 

Bayesian estimation of community size and overlap from random subsamples

Dear Dr Larremore,

I am pleased to inform you that your manuscript has been formally accepted for publication in PLOS Computational Biology. Your manuscript is now with our production department and you will be notified of the publication date in due course.

With kind regards,

Agnes Pap
